# Hipposeq: a comprehensive RNA-seq database of gene expression in hippocampal principal neurons

**Mark S Cembrowski, Lihua Wang, Ken Sugino, Brenda C Shields, Nelson Spruston***

Janelia Research Campus, Howard Hughes Medical Institute, Ashburn, United States

**Abstract** Clarifying gene expression in narrowly defined neuronal populations can provide insight into cellular identity, computation, and functionality. Here, we used next-generation RNA sequencing (RNA-seq) to produce a quantitative, whole genome characterization of gene expression for the major excitatory neuronal classes of the hippocampus; namely, granule cells and mossy cells of the dentate gyrus, and pyramidal cells of areas CA3, CA2, and CA1. Moreover, for the canonical cell classes of the trisynaptic loop, we profiled transcriptomes at both dorsal and ventral poles, producing a cell-class- and region-specific transcriptional description for these populations. This dataset clarifies the transcriptional properties and identities of lesser-known cell classes, and moreover reveals unexpected variation in the trisynaptic loop across the dorsal-ventral axis. We have created a public resource, Hipposeq (http://hipposeq.janelia.org), which provides analysis and visualization of these data and will act as a roadmap relating molecules to cells, circuits, and computation in the hippocampus.

**\*For correspondence:**
sprustonn@janelia.hhmi.org

**Competing interests:** The authors declare that no competing interests exist.

## Introduction

Gene expression profiling can be a powerful tool to understand the functionality and organization of cells and networks. For example, by relating the specific enriched or depleted genes to their corresponding ontologies in a given population, functional hypotheses can be generated at the intrinsic level (e.g., voltage-gated channel subunits) and network level (e.g., ligand-receptor interactions). A different approach, agnostic to functional correlates of genes, can be taken by using gene expression profiles as a means to genetically delineate different populations of cells, either across classes or within a given class. In this way, gene expression profiling simultaneously clarifies complementary aspects of molecular, cellular, and circuit properties of cells.

Transcriptional profiling in the mouse brain is becoming a powerful tool in neuroscience, owing to a host of complementary innovations and technologies. A variety of transgenic mice have emerged over the last decade (*Gong et al., 2003*; *2007*; *Taniguchi et al., 2011*), which enables access to genetically defined populations of neurons, and a variety of techniques now exist for purifying labeled cells from surrounding tissue (*Okaty et al., 2011a*) to obtain cell-class-specific transcriptomes (*Okaty et al., 2011b*). Although large-scale, quantitative gene expression profiling across neurons has typically been performed by microarray (*Belgard et al., 2011*; *Siegert et al., 2012*; *Sugino et al., 2006*), more recently the technically superior RNA-seq (*Shin et al., 2014*) is finding application in the neurosciences (*Cembrowski et al., 2016*; *Zeisel et al., 2015*; *Zhang et al., 2014*). Complementing these quantitative profiling methods is the mouse Allen Brain Atlas (ABA) (*Lein et al., 2007*), providing histological information from in situ hybridization (ISH) assays.

**eLife digest** Both mouse and human brains are made up of many millions of cells called neurons that are interconnected to form circuits. These neurons are not all the same, because different classes of neurons express different complements of genes. Neurons that express similar genes tend to look and act alike, whereas neurons that express different genes tend to be dissimilar.

Cembrowski et al. have used a technique called next-generation RNA sequencing (RNA-seq) to determine which genes are expressed in groups of neurons that represent the main cell types found in a part of the brain called the hippocampus. This brain region is important for memory, and was chosen because the location and appearance of the main cell types in the hippocampus were already well understood.

The approach revealed that the main types of neurons in the mouse hippocampus are all very different from each other in terms of gene expression, and that even neurons of the same type can exhibit large differences across the hippocampus. Cembrowski et al. created a website that will allow other researchers to easily navigate, analyze, and visualize gene expression data in these populations of neurons.

Future work could next make use of recent technological advances to analyze gene expression in individual neurons, rather than groups of cells, to provide an even more detailed picture. It is also hoped that understanding the differences in gene expression will guide examination of how the hippocampus contributes to memory and what goes wrong in diseases that affect this region of the brain.

A combination of these techniques has been previously applied to study principal cells in the hippocampus. Microarray work has been used to study CA1 pyramidal cells (*Kamme et al., 2003*; *Sugino et al., 2006*) as well as cells of the trisynaptic loop (*Deguchi et al., 2011*; *Greene et al., 2009*; *Lein et al., 2004*; *Nakamura et al., 2011*; *Zhao et al., 2001*). Mining of the ABA has revealed molecularly defined subregions in multiple principal cell classes (*Dong et al., 2009*; *Fanselow and Dong, 2010*; *Thompson et al., 2008*). Most recently, RNA-seq work has been used to study CA1 pyramidal cells (*Cembrowski et al., 2016*; *Zeisel et al., 2015*). This work has helped to identify genetic differences within and across regions and as well revealed differences within canonical neuronal populations.

Despite this extensive investigation, many aspects of the hippocampal transcriptome remain unresolved or warrant further investigation. Previous transcriptional profiling has predominantly focused on CA1 and CA3 pyramidal cells; markedly less work has examined DG granule cells and CA2 pyramidal cells (but see *Fanselow and Dong, 2010*; *Lein et al., 2004*; *2005*) and no profiling has been performed for DG mossy cells. Additionally, recent work has suggested that DG granule cells may be a heterogeneous population along the dorsal-ventral axis (*Fanselow and Dong, 2010*), but this has not been systematically investigated. Perhaps most importantly, the technical superiority of RNA-seq may reveal governing organizational rules that may have not been resolved with ISH or microarray (*Cembrowski et al., 2016*), suggesting that a systematic RNA-seq based approach may provide unparalleled insight into the transcriptional organization of the hippocampus.

Here, we manually purified labeled excitatory cell populations from microdissected hippocampal regions, and used RNA-seq in combination with histological information from ABA to characterize gene expression quantitatively and histologically. This approach enabled analysis of hippocampal gene expression in a cell-class- and region-specific manner. We use this approach to examine both previously characterized and novel transcriptomes, and to understand the organizational schemes of gene expression within and across neuronal populations. These data and analysis tools are publicly available (http://hipposeq.janelia.org), enabling users to examine gene expression at multiple levels of granularity in the hippocampus and providing a molecular blueprint to predict and investigate phenotypes at cellular, systems, and behavioral levels.

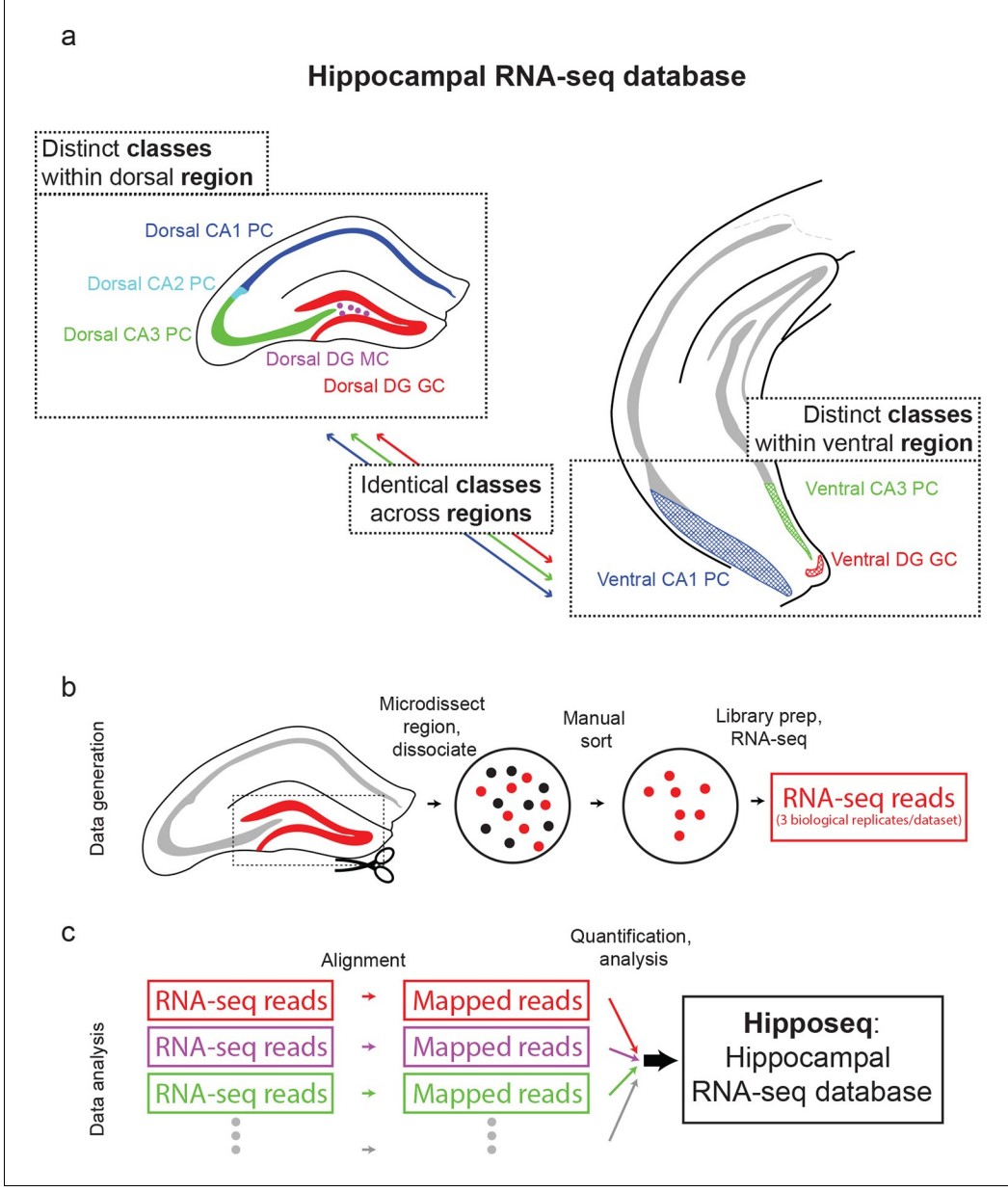

**Figure 1.** Generation of hippocampal RNA-seq database. (**a**) Datasets included in the hippocampal RNA-seq characterization. Note, operationally, cell 'class' refers to gross cell type and 'region' refers to dorsal vs. ventral location. (**b**) Protocol underlying the generation of raw RNA-seq data. In a transgenic line in which cells of interest were fluorescently labeled (left), the region of interest was microdissected (dashed box). The isolated region was then dissociated, and labeled neurons were manually purified (middle). RNA-seq data was generated from the purified cells. (**c**) Protocol underlying the processing of RNA-seq data. Raw reads were aligned, and then expression was quantified and statistically analyzed.

The following figure supplements are available for figure 1:

**Figure supplement 1.** Transgenic lines used to create cell-class- and region-specific transcriptomes.

**Figure supplement 2.** Reproducibility and purity of RNA-seq data.

**Figure supplement 3.** Reproducibility of RNA-seq quantification and differential expression.

## Results

### Generating a cell-class- and region-specific RNA-seq database for the hippocampus

The hippocampus is grossly comprised of five excitatory cell populations; namely, granule and mossy cells of the dentate gyrus (DG), and pyramidal cells of CA3, CA2, and CA1. We sought to obtain and analyze transcriptomes for each of these five excitatory populations, which we operationally refer to as five distinct cell 'classes' for the remainder of the manuscript. Additionally, following recent work illustrating prominent dorsal-ventral differences within multiple canonical cell classes in the hippo-campus (*Cembrowski et al., 2016*; *Dong et al., 2009*; *Fanselow and Dong, 2010*; *Thompson et al., 2008*), we endeavored to profile the excitatory cells comprising the trisynaptic loop at the dorsal and ventral poles of the hippocampus; operationally, hereafter we will refer to cell classes at opposite poles to be from distinct 'regions'. Thus, in total, we aimed to profile eight dis-tinct excitatory neuronal populations based upon cell-class and region specificity (*Figure 1a*).

To transcriptionally profile each of the eight populations (*Figure 1b,c*), we first identified trans-genic mouse lines that would allow for class and region specificity when combining local microdis-sections with fluorescence-based purification (see Materials and methods; *Figure 1—figure supplement 1*). We then microdissected the region of interest from the corresponding transgenic animal; this tissue was subsequently dissociated and the fluorescently labeled cells were purified by manual selection (112 ± 6 cells per biological replicate, mean ± SEM, n = 24 replicates) (*Hempel et al., 2007*). The sorted sample underwent library preparation and sequencing, the result-ing raw RNA-seq reads were aligned, and expression was quantified and analyzed across samples (see Materials and methods).

To assess reproducibility, three biological replicates were ascertained for each dataset. Replicate datasets, corresponding to the same class-region pair, were well correlated with each other (r = 0.98 ± 0.02, mean ± SD, Pearson's correlation coefficient; *Figure 1—figure supplement 2a,b*), and each replicate was devoid of marker gene cohorts associated with interneurons and non-neuronal cells (*Figure 1—figure supplement 2d*). Thus, our obtained transcriptomes were internally consistent and cell-class specific, ensuring the integrity of our dataset.

### A quantitative overview of hippocampal gene expression

We began by exploring the gross relationships of hippocampal transcriptomes. Using hierarchical clustering (see Materials and methods; *Figure 2a*) we found the initial bifurcation corresponded to a divide between granule cells and non-granule cells, consistent with previous microarray (*Greene et al., 2009*) and ISH work (*Thompson et al., 2008*). The second broad division of the den-drogram partitioned mossy cells from pyramidal cells and the final bifurcation in each limb corre-sponded to dorsal-ventral differences in each cell class, although the degree of within-class similarity was frequently comparable to across-class similarity (*Cembrowski et al., 2016*).

We next cross-validated our RNA-seq hits with ABA ISH data (see Materials and methods). From RNA-seq, many marker genes could be identified that corresponded to specific dendrogram bifurca-tions, both across broad hippocampal populations (*Figure 2a*, left) as well as within cell classes across regions (*Figure 2a*, right). Importantly, these RNA-seq hits gave good agreement with ABA histological data, correctly predicting the enriched populations in ~81% of cases (124/153 genes where coronal ISH images were available, *Figure 2b*, *Supplementary file 1*; see Materials and methods).

The consistency of RNA-seq with existing ISH data indicates that the two datasets can be used in conjunction to study spatial patterns of gene expression and delineate genetic boundaries across excitatory cell classes in the hippocampus. Complementing this, the quantitative whole-genome nature of RNA-seq enables well-principled numerical insight into the extent and properties of gene enrichment. For the remainder of the manuscript, we leverage these advantages to first examine individual cell classes, and then subsequently elucidate transcriptomes across cell classes and regions of the hippocampus.

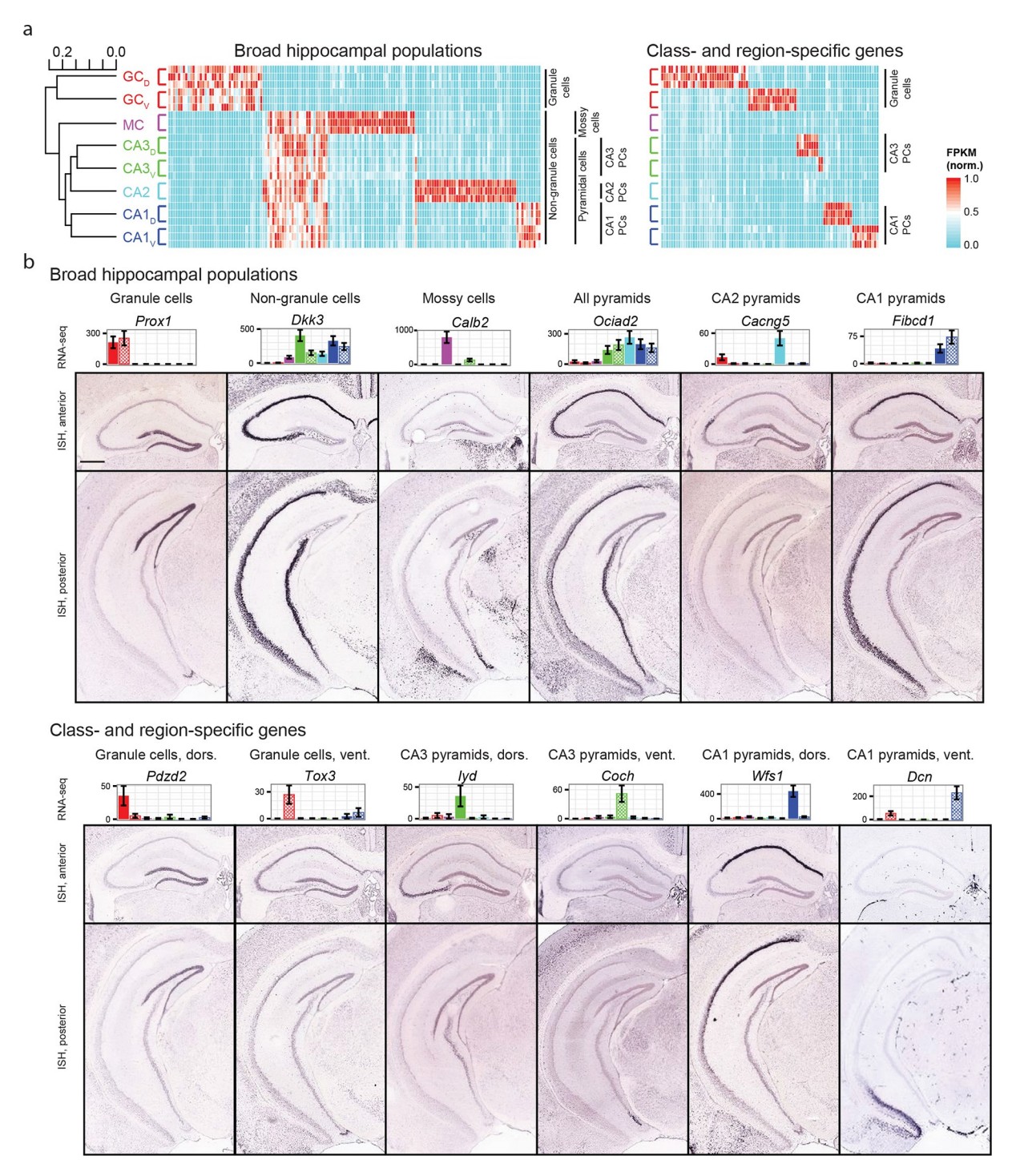

**Figure 2.** Gene expression in the hippocampus exhibits a variety of cell population- and region-specific expression. (a) Left: the hierarchical structure of gene expression in the hippocampus calculated by agglomerative clustering. Middle and right: Expression across replicates for marker genes associated with broad hippocampal populations (middle) or specific cell classes and regions (right). Marker genes were selected based upon two-fold enrichment in all replicates in the target population(s) relative to all other replicates (see Materials and methods). FPKM values displayed in the heat map were normalized on a gene-by-gene (i.e., column-by-column) basis by the highest expressing sample for each gene. (b) Confirmation of gene expression profiles by ISH. In corresponding bar plots, RNA-seq FPKM values for each class/region dataset are displayed, with coloring adhering to the conventions of *Figure 1a* and fill vs. crosshatch indicating dorsal vs. ventral datasets. Scale bar: 500 μm.

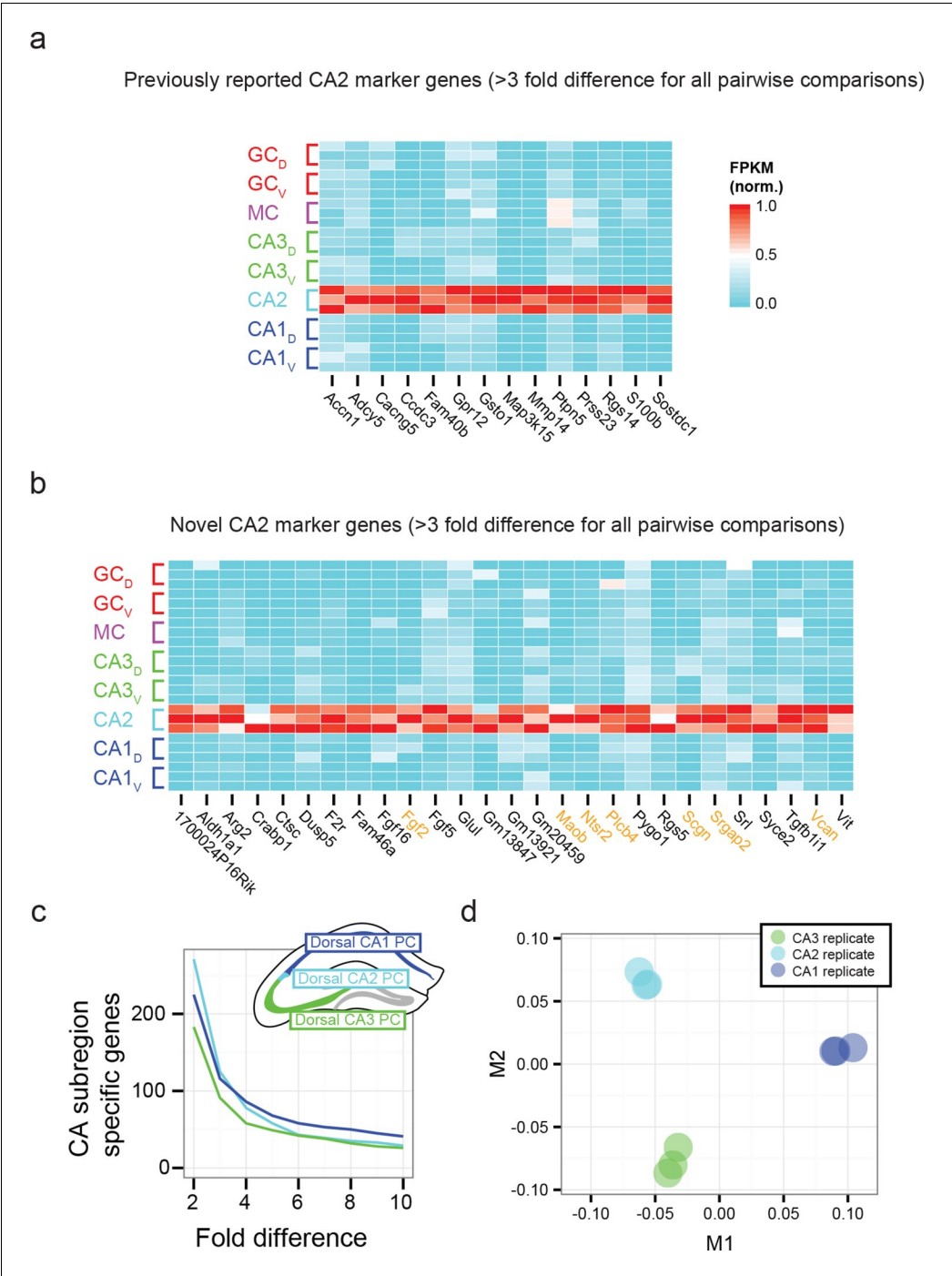

**Figure 3.** Gene expression properties of CA2 pyramidal cells. (a) Heat map of replicate FPKM values for previously identified CA2 marker genes. (b) Heat map of replicate FPKM values for novel CA2 marker genes identified by RNA-seq. Orange indicates genes with previously characterized neuronal relevance. (c) The number of CA3-, CA2-, and CA1-specific genes, when restricting comparisons to solely these three cell populations in the dorsal hippocampus. A gene is denoted as X-fold enriched in a given CA region if it the average FPKM value is at least X-fold greater than the other CA regions. (d) Multidimensional scaling demonstrating the distance between CA3, CA2, and CA1 pyramidal cells.

The following figure supplement is available for figure 3:

**Figure supplement 1.** Recapitulation and extension of previous CA2 marker gene results.

## CA2 transcriptional identity

To examine the extent to which our RNA-seq dataset both recapitulated and expanded upon previous work, we searched for CA2-specific marker genes in our RNA-seq dataset. This search identified 41 genes with >3-fold enrichment in CA2 relative to all other populations, using relatively conservative search parameters (*Figure 3a,b*; see Materials and methods). We compared these genes against previously known CA2-enriched genes from both literature (*Dudek et al., 2016*) and ABA mining (*Lein et al., 2007*). Notably, although some of our retrieved genes were previously identified as enriched in CA2 (*Figure 3a*), the majority of discovered genes were novel hits (66%, n=27/41; *Figure 3b*). Thus, our dataset recapitulated previous findings, but moreover revealed a host of previously unidentified genes with greater CA2 specificity (*Figure 3—figure supplement 1*), directly demonstrating the utility of RNA-seq relative to previous methodologies. Many novel genes were associated with functionally relevant neuronal ontologies, including cell adhesion and axon guidance (*Srgap2, Vcan*), neuropeptide signaling (*Ntsr2*), and calcium binding (*Scgn*) (genes highlighted in orange, *Figure 3b*).

Although area CA2 shares some similarities with neighboring CA3 and CA1 regions, CA2 also exhibits features unique among these principal cells. Consequently, the extent to which CA2 pyramidal cells embody their own unique characteristics versus sharing properties with CA1 and/or CA3 is a subject of ongoing research (*Dudek et al., 2016*), which can be directly and comprehensively addressed by transcriptome comparisons. Analyzing gene expression in dorsal CA3, CA2, and CA1, we found each cell population had a similar number of enriched genes (*Figure 3c*). Complementing this, applying multidimensional scaling to visualize the distances between CA3, CA2, and CA1 (see Materials and methods), we found that the three regions were approximately equidistant (*Figure 3d*). These results illustrated that CA2 is largely its own distinct region, rather than being a weighted combination of CA3 and CA1 features; i.e., the physical intermediacy of CA2 did not correlate with transcriptional intermediacy.

## Mossy cell transcriptional identity and variability

We also investigated mossy cells, a relatively uncharacterized excitatory cell population found within the hilus of the dentate gyrus. As with CA2, we first investigated the extent to which mossy cells exhibited enriched genes relative to all other hippocampal excitatory neurons. Previous work has found one gene enriched in mossy cells (*Calb2*) (*Fujise et al., 1998*), which was recapitulated by our analysis (*Figure 4a*); in addition we identified 59 mossy cell-enriched genes in the hippocampus (*Figure 4b*). Many of these genes play roles in neuronally relevant ontologies (genes highlighted in orange, *Figure 4b*), including cellular adhesion and axon guidance (*Cntn6, Ephb6*), calcium signaling (*Hpcal1*), ligand-receptor signaling (*Drd2, Gal, Glp1r, Grm8, Nmb)*, and regulation of transcription (*Prrx1*).

Cross-validating these genes in the ABA, we found excellent agreement between mossy cell-enriched genes from RNA-seq and expression in the hilar cells (95%; n=36/38 agreement where coronal images available, *Supplementary file 2*; see Materials and methods). Interestingly, although some genes seemed to be expressed relatively uniformly across the long axis (e.g., *Csf2rb2*, *Figure 4c*), many genes seemed to be enriched at specific locations along the long axis. For example, *Nmb* and *Thbs2* exhibited expression near the dorsal pole of the hippocampus but lacked expression at the ventral horn of the hippocampus. Conversely, *Calb2* and *Tm4sf1* exhibited expression concentrated near the ventral pole of the hippocampus. In addition to this differential labeling across the hippocampus, differences were also seen in the labeling density at corresponding enriched regions (e.g., cf. *Nmb* with *Thbs2* dorsally and *Calb2* with *Csf2rb2* ventrally), suggesting that mossy cells are a transcriptionally heterogeneous population of cells.

## Granule cell transcriptional identity and variability

Although significant work has been done examining dorsal-ventral differences in CA3 and CA1 (*Cembrowski et al., 2016*; *Thompson et al., 2008*), differences in dentate gyrus granule cells have received relatively little attention. Previous work (*Fanselow and Dong, 2010*) has suggested that domains specified by the dorsal marker gene *Lct* and the ventral marker *Trhr* may correspond to tripartite molecular divisions of the dentate gyrus (*Figure 5a*): here, *Lct* and *Trhr* expression respectively represent the dorsal and ventral divisions, whereas the intermediate domain is characterized

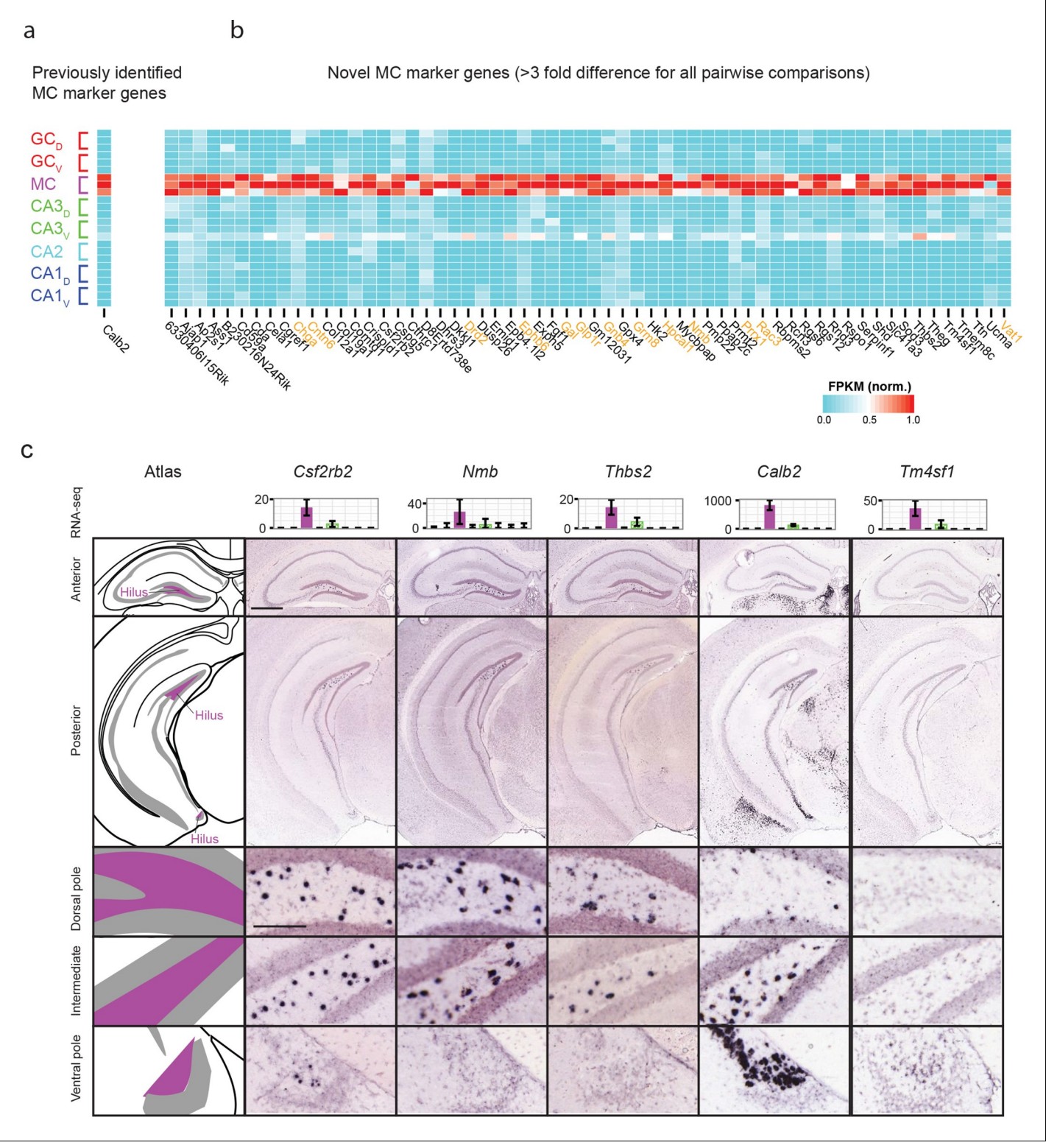

**Figure 4.** Gene expression properties of hilar mossy cells. (**a**) Heat map of replicate FPKM values for the previously identified mossy cell marker gene *Calb2*. (**b**) Heat map of replicate FPKM values for novel mossy cell marker genes identified by RNA-seq. Orange indicates genes with previously characterized neuronal relevance. (**c**) ISH profiles (bottom) for marker genes identified by RNA-seq (top). Scale bar, overview: 500 μm; expanded: 100 μm.

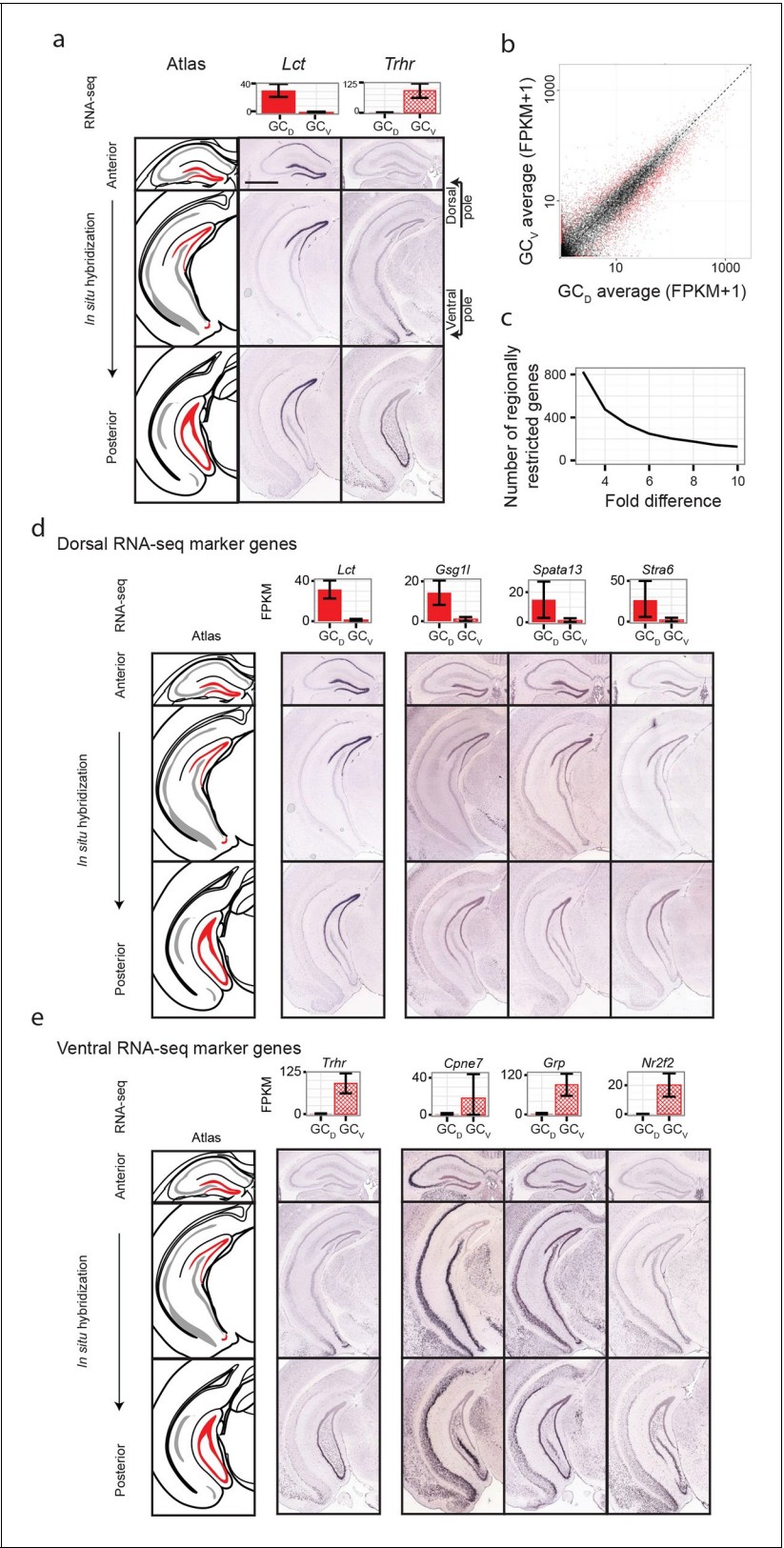

**Figure 5.** Dorsal-ventral differences in dentate gyrus granule cells. (**a**) RNA-seq (top) and ISH (bottom) profiles of *Lct* and *Trhr*, two previously identified marker genes respectively enriched in dorsal and ventral granule cells. Scale bar: 500 μm. (**b**) FPKM scatterplot of average dorsal and ventral GC transcriptomes. Data points represent individual genes, with genes highlighted in red indicating differential expression. (**c**) Number of genes enriched
*Figure 5 continued on next page*

*Figure 5 continued*

at the poles of DG as a function of fold change. (**d**) Example ISH profiles (bottom) of dorsal GC marker genes obtained by RNA-seq (top). (**e**) As in (**d**) but for ventral GC marker genes.

The following figure supplement is available for figure 5:

**Figure supplement 1.** Dorsal-ventral differences in dentate gyrus granule cells.

by weak expression of both genes. Importantly, our RNA-seq work recapitulated both of these marker genes, suggesting that our data could be used to quantitatively explore the degree and patterns of granule cell heterogeneity.

We first used our RNA-seq dataset to examine gene expression for granule cells at the two poles of the dentate gyrus. Notably, hundreds of genes were differentially expressed between these two poles (*Figure 5b*), and corresponded to large fold changes (*Figure 5c*). Many of these genes were involved in neuronally relevant functions and were cross-validated by ISH (58%, n=33/57 agreement where coronal images available, *Figure 5—figure supplement 1*, *Supplementary file 3*; see Materials and methods).

Given the agreement between RNA-seq and ISH, we used the ABA to investigate genetic domains in granule cells. The genetic domain specified by *Lct* was recapitulated by multiple dorsal marker genes (*Figure 5—figure supplement 1b*), including *Gsg1l, Spata13,* and *Stra6* (*Figure 5d*). Conversely, little agreement was seen between ventral marker genes: the domain specified by *Trhr* was found to differ from the patterns observed for other markers (e.g., *Cpne7, Grp,* and *Nr2f2*; *Figure 5e*, *Figure 5—figure supplement 1c*). These findings indicate that the genetic boundary defined by *Lct* may correspond to a transcriptionally well-defined subpopulation, but an equally well-defined ventral subpopulation does not appear to be present. Despite this, all novel marker genes validated by ISH appeared to be expressed in gradients across the long axis (*Figure 5—figure supplement 1b,c*; we did not find genes that were selectively expressed in either the upper or lower blades of DG granule cells), suggesting that granule cell transcriptional identity exists in a continuous spectrum in this axis.

## Dorsal-ventral differences independent of cell class

The granule cell marker genes *Lct* and *Trhr* were previously shown to be enriched in other class-region populations; namely, *Lct* is expressed in dorsal CA1 and *Trhr* is expressed in ventral CA3 (*Cembrowski et al., 2016*; *Dong et al., 2009*; *Thompson et al., 2008*) (see also *Figure 5a*). This raises the intriguing possibility that there exist genes that are enriched in a region- but not class-specific manner; i.e., genes enriched dorsally or ventrally across multiple cell classes. We next analyzed this the context of the dorsal versus ventral cell classes of the trisynaptic loop.

We first identified the number of expressed genes that were >2 fold enriched between poles on a class-by-class basis (top values, *Figure 6a*). From here, we searched for genes that were associated with enrichment at the same pole across multiple class comparisons; e.g., the genes *Cadm2* and *Mgll* were found to be >2 fold dorsally enriched in every dorsal-ventral comparison, whereas *Resp18* and *Efnb2* were found to be ventrally enriched (*Figure 6b*; corroborated by ABA, *Figure 6d*). In general, many genes were found that obeyed region-specific enrichment across multiple cell classes (*Figure 6c*). To compare this empirically determined number of region-enriched genes relative to the number expected by chance, we calculated the total number of genes that were expressed in each cell class/region combination (defined as the number of genes with $FPKM_{avg}>10$; *Figure 6a*), and compared this experimental data to a null model where gene names were drawn at random from the list of expressed genes (see Materials and methods) (*Figure 6c*). Interestingly, in almost every possible comparison (n=5/6 pairwise enrichment combinations and 2/2 triplicate enrichment combinations), the number of enriched genes that were shared across cell classes in a region-specific manner were significantly greater than that expected by chance.

We examined the ontologies associated with the 37 genes that were enriched across all dorsal-ventral comparisons (i.e., the n=12 dorsally and 25 ventrally enriched genes from the triplicate comparisons of *Figure 6c*). Although the enriched genes spanned a variety of ontologies, many genes that emerged as being region- but not class-enriched corresponded to cellular adhesion and axon

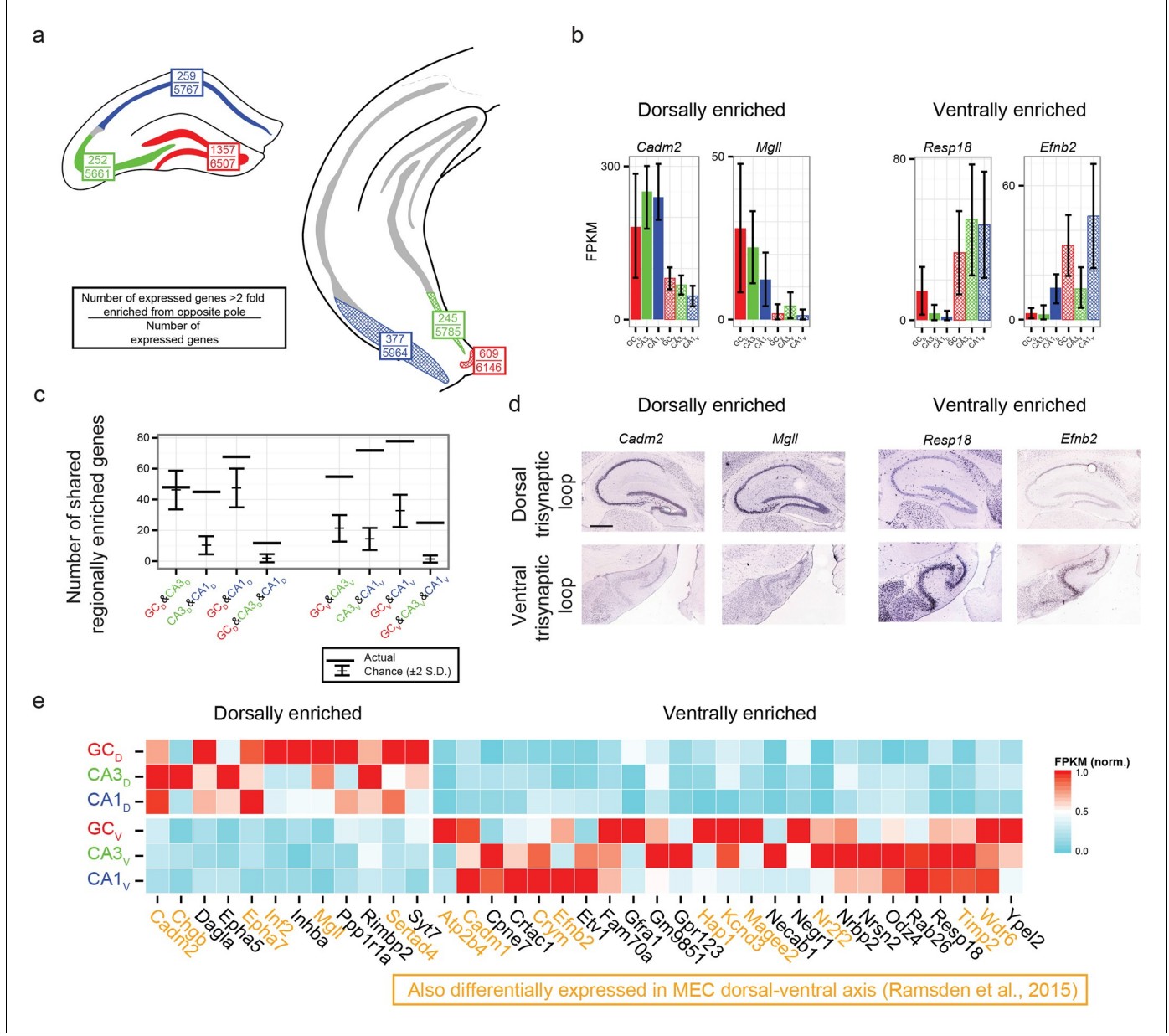

**Figure 6.** Regionally enriched genes invariant to principal cell class. (**a**) For each trisynaptic loop dataset, the number of enriched genes when comparing to the same cell class at the opposite pole (>2-fold difference), as well as the total number of expressed genes for each dataset (FPKM_MIN>10), are shown (top and bottom values respectively). (**b**) Example genes enriched in a region-, but not cell-class-, specific manner. (**c**) The number of dorsally and ventrally enriched genes shared across cell classes. Both the observed RNA-seq data (horizontal lines) and null distribution (mean ± 2SD) are shown. (**d**) Sagittal ISH profiles of the example region-enriched genes. Scale bar: 500 μm. (**e**) Heat map of all genes found enriched in a region-specific manner across the trisynaptic loop (n=37; null distribution predicts 6.0 ± 7.1 (mean ± 2SD), p<1e-6). Orange text: genes also identified as differentially expressed in medial entorhinal cortex (MEC) (*Ramsden et al., 2015*).

The following figure supplement is available for figure 6:

**Figure supplement 1.** Dorsal and ventral genes enriched across hippocampus and MEC.

guidance; for example, the cellular adhesion molecules *Cadm1, Cadm2*, the ephrins/receptors *Epha5, Epha7, Efnb2,* as well as *Dagla, Odz4, Timp2* and *Negr1* (*Figure 6e*).

Given the abundance of genes expressed in a region-specific manner in the hippocampus, we next examined whether these region-specific genes would predict similar patterning outside of the

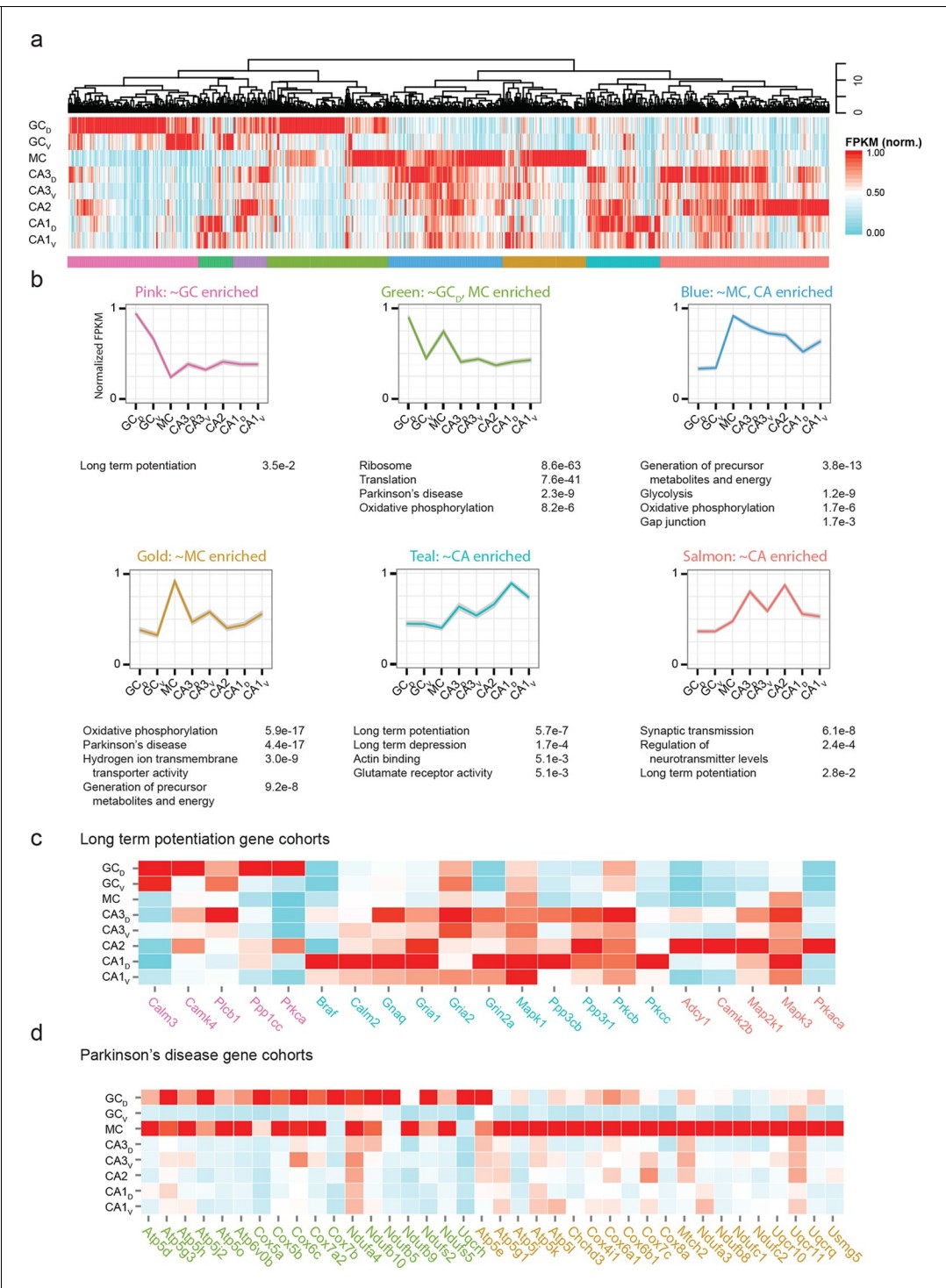

**Figure 7.** Weighted gene co-expression network analysis (WGCNA) of hippocampal excitatory neuron transcriptomes. (a) Top and middle: hierarchical clustering and normalized expression of the 1000 most variable genes, respectively. Bottom: colors denoting the modules obtained from WGCNA. (b) Six modules obtained from a, with the average expression shown and significantly enriched terms highlighted. Each module is named according to the gross overall expression profile across datasets. (c) Genes associated with long-term potentiation significantly enriched in modules. (d) As in c, but with genes associated with Parkinson's disease.

hippocampus. Recently, RNA-seq has been conducted on the dorsal and ventral poles of the medial entorhinal cortex (MEC) (*Ramsden et al., 2015*), providing a direct comparison with our data. Strikingly, of the 37 genes identified as regionally enriched in all principal cells of the trisynaptic loop, 43% (n=16/37) were identified as differentially expressed in the dorsal-ventral axis of MEC with an identical directionality (*Figure 6e*, orange text). Examining the spatial patterns of gene expression in the MEC in sagittal sections, we found that these regional-specific genes exhibited a broad range of expression profiles across the dorsal-ventral axis, attenuating in labeling density and/or intensity on a gene-by-gene basis (*Figure 6—figure supplement 1*). Attenuation was also generally not constrained to a fixed cell class: although some genes exhibited expression restricted to single lamina (e.g., *Inf2, Etv1*), other genes were more broadly expressed (e.g., *Efnb2* across two laminae; *Cadm2, Crym, Hap1,* and *Odz4* across 3+ laminae).

## Enriched gene modules in hippocampal subpopulations

The preceding work considered cell classes and regions determined *a priori* to analysis. To complement this, we next used a wholly data-driven approach to analyze the hippocampal transcriptome through Weighted Gene Co-expression Network Analysis (WGCNA; see Materials and methods) (*Zhang and Horvath, 2005*). This method identifies highly correlated expression of gene modules across subsets of samples. We used the top 1000 most variable genes from the full hippocampal dataset, and using WGCNA, identified eight gene modules that were enriched in various subpopulations of hippocampal excitatory neurons (*Figure 7a,b*; see Materials and methods).

The functional implications of these modules were then examined by using DAVID (*Huang et al., 2009a*; *2009b*) (see Materials and methods) to identify statistically significant Gene Ontology and KEGG Pathway terms (*Kanehisa and Goto, 2000*; *Kanehisa et al., 2014*). Interestingly, although by definition the genes present within a given module were not shared across modules, many associated ontologies and pathways were common between modules. For example, the KEGG annotation 'Long Term Potentiation' was enriched for distinct modules associated with DG granule cells, CA1 pyramidal cells, and CA2/3 pyramidal cells (*Figure 7c*), illustrating that specific genes that underlie LTP vary between cell classes despite all cell classes expressing genes related to LTP. Similarly, this approach also enabled us to identify specific modules with disease annotations; for example, both mossy cells and dorsal dentate gyrus granule cells were enriched for genes associated with disease terms (e.g., Parkinson's; *Figure 7d*).

## Discussion

Here, we have used RNA-seq in conjunction with ABA in situ hybridization to transcriptionally profile populations of excitatory neurons in the hippocampus. Employing a combination of fluorescent cell sorting and regional microdissection, we obtained cell-class- and region-specific transcriptomes from eight distinct populations of hippocampal neurons. The transcriptional data obtained here provide insight into multiple complementary properties of the hippocampus. For example, the greatly expanded number of marker genes fosters predictions for the role of genes and cell classes in physiology and disease, the complement of RNA-seq and ISH reveals the spatial relationships within and across cell classes of the hippocampus, and the dorsal versus ventral profiling demonstrates the surprising finding that gene expression can be regionally enriched invariant to distinct cell classes. We have made these data publicly available, augmented with user-friendly interactive analysis and visualization tools (http://hipposeq.janelia.org), providing readily available and customizable RNA-seq exploration to the hippocampal community.

### Identification and utility of novel marker genes

A major result of our RNA-seq analysis was the identification of region- and/or class-specific marker genes. Our profiling recapitulated many marker genes known *a priori*, but in addition, demonstrated an abundance of previously unappreciated marker genes. Consequently, our work greatly expands the total number of marker genes and enables several complementary lines of inquiry based upon these findings. First, the marker genes uncovered here serve as candidates for obtaining genetic access to well-defined populations of neurons. Due to the inherently quantitative nature of RNA-seq, the strength and specificity of candidate genes can be evaluated by examining expression in both on-target and off-target populations; promoters associated with genes that are sufficiently restricted

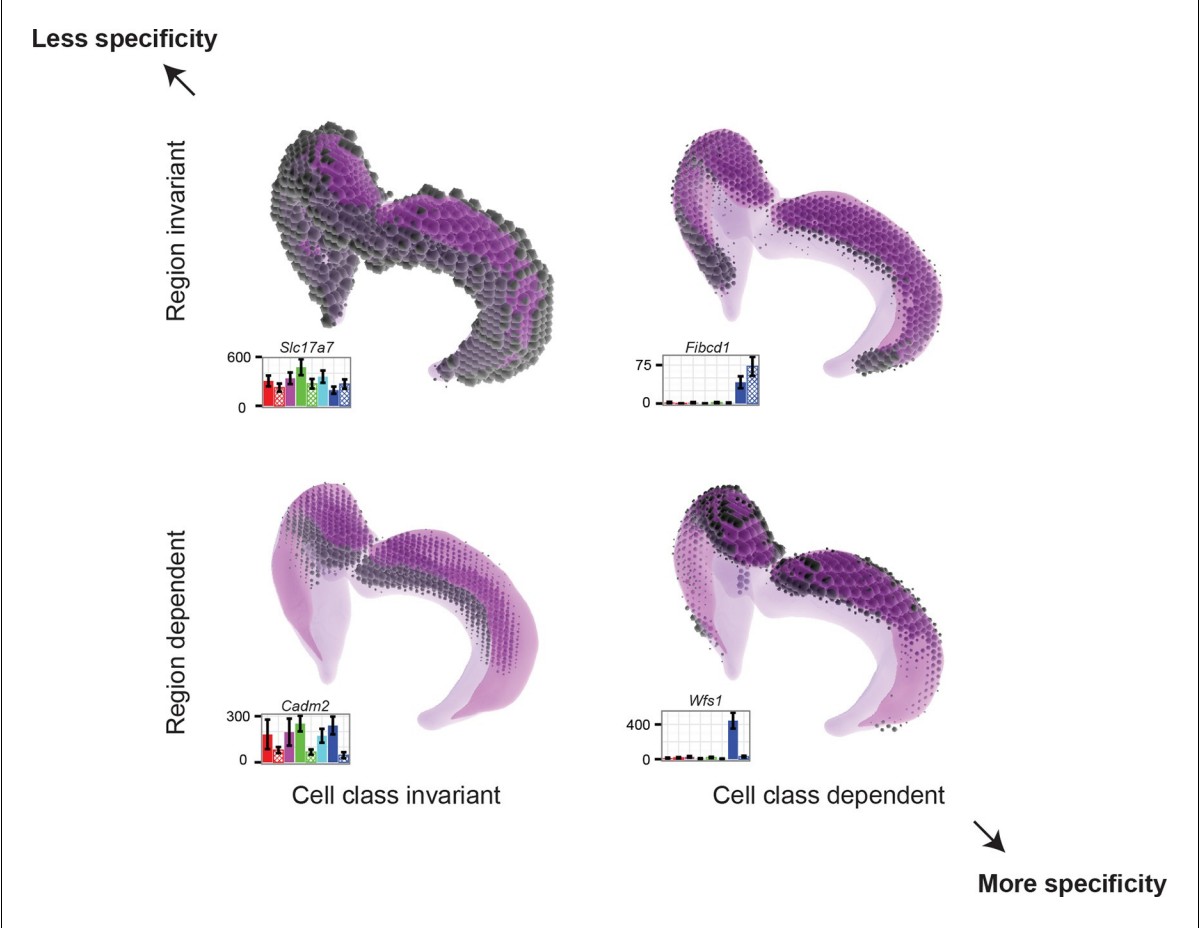

**Figure 8.** Gene expression patterns of excitatory hippocampal neurons. Genes can be expressed in relatively similar abundances across the hippocampus (*Slc17a7*; upper left), vary in a cell-class-specific (*Fibcd1*; upper right) or region-specific manner (*Cadm2*; lower left), or vary in both cell-class- and region-specific manners simultaneously (*Wfs1*; lower right). In each panel, light magenta denotes the spatial extent of the hippocampus, dark magenta illustrates the CA1 region in particular, and the dots indicate the location and intensity of labeling from ISH. RNA-seq profiling results are provided for each gene. Images from the Allen Brain Explorer v2.

in expression can be employed for designing transgenic animals or viruses to enable genetic access. We emphasize that the Cre lines used here for labeling cells for transcriptional profiling (*Figure 1— figure supplement 1*) may already provide sufficient genetic access to many excitatory subpopulations of the hippocampus; thus, this work offers both existing and new ways to selectively target and manipulate specific populations of neurons. Second, many of the marker genes identified here can be tied to specific functionality (e.g., *Figure 3b*, *4b*, *Figure 5—figure supplement 1a*), allowing novel marker genes to be used for hypothesis generation. These hypotheses can be investigated through a variety of perturbations (e.g., siRNA, knockouts, or CRISPR-Cas gene editing) combined with other experiments (e.g., physiology, behavior). Similarly, many marker genes identified here are annotated as specific disease-related genes (e.g., *Figure 7b,d*), which can help to reveal both the molecules and cell classes to examine in pathological conditions and disease models.

## Cell-class- and region-specific gene expression in the trisynaptic loop
Our work here identified two components of transcriptional variability in the cells of the trisynaptic loop; namely, differences across different neuronal populations (across-class) and differences across the dorsal-ventral axis (across-region). On a gene-by-gene basis, these two organizational principles could be observed either individually or simultaneously (*Figure 8*). Transcriptional differences across classes of the trisynaptic loop have been identified and appreciated for some time (e.g., *Greene et al., 2009*; *Lein et al., 2004*). Indeed, it is likely that these transcriptional differences are

fundamental in producing across-class variability in morphology, physiology, and connectivity, ultimately underlying the diverse and distinct roles that the different cell classes are believed to play in hippocampal processing (*Mizuseki et al., 2012*; *Neunuebel and Knierim, 2014*). Across-region differences, at a cell-class-specific level, have recently been identified to be present in CA3 and CA1 pyramidal cell populations (*Cembrowski et al., 2016*; *Thompson et al., 2008*). Here, we show that granule cells of the dentate gyrus also adhere to this rule, with marked differences present in the transcriptomes between dorsal and ventral poles (*Figure 5b,c*). Indeed, the transcriptional distance between dorsal and ventral granule cells is similar to that between different populations of cells (*Figure 2a*), a finding similar to that found between CA3 and CA1 pyramidal cells (*Figure 2a*) (*Cembrowski et al., 2016*).

Finding genes that are regionally enriched across cell classes (*Figure 6*) is surprising and warrants further investigation. The fact that many of these genes are involved in cell adhesion and axon guidance, in conjunction with the observation that they are enriched along the dorsal-ventral axis in multiple areas of the brain, suggests that they may generally be used for maintaining polarity in the mature brain. During development, gradients of gene expression are used for proper patterning of neural circuits (*Sansom and Livesey, 2009*); in a similar fashion, these genes may reflect the mature counterpart that actively maintains spatial identity.

## Transcriptional variability in excitatory cells of the hippocampus

A central goal of neuroscience is to disentangle and understand the vast complexity of neuronal populations, both within and between cell classes. Next-generation RNA-seq provides a comprehensive means of clarifying cellular identities in the hippocampus and complements other gene expression analyses (*Dong et al., 2009*; *Fanselow and Dong, 2010*; *Lein et al., 2007*; *Thompson et al., 2008*). Our work here furthers understanding of across-class differences, but also emphasizes the high degree of transcriptional variability that can be present within a given population (e.g., across the dorsal-ventral axis of the hippocampus). Pyramidal cells in CA1 and CA3 (*Cembrowski et al., 2016*; *Thompson et al., 2008*), as well as both mossy cells (*Figure 4*) and granule cells (*Figure 5*) of the dentate gyrus, appear to exhibit a high degree of heterogeneity across the long axis. Notably, this within-class, dorsal-ventral heterogeneity can exhibit different organizational principles; for example, CA3 pyramidal cells have been shown to conform to discrete subpopulations (*Thompson et al., 2008*), whereas CA1 pyramidal cells (*Cembrowski et al., 2016*), DG mossy cells (*Figure 4*), and DG granule cells (*Figure 5*) do not exhibit clear subdomain organization.

It is important to emphasize that our results, although illustrating a high degree of transcriptional heterogeneity in principal cells of the hippocampus, likely underestimate the total amount of variability for several reasons. First, comparing the dorsal and ventral poles may miss differences present along other dimensions, as well as differences present at spatially intermediate locations. Second, our population-level approach may miss heterogeneity that is present at a subpopulation or single-cell level, including transcriptional signatures associated with specialized but sparse excitatory cell classes (e.g., radiatum giant cells (*Gulyas et al., 1998*), semilunar granule cells (*Williams et al., 2007*), and CA3 granule cells (*Szabadics et al., 2010*)). Finally, transcriptional properties may vary in ways other than geographical location; for example, differential gene expression for cells of the same region that target different downstream locations (*Cembrowski et al., 2016*; *Sorensen et al., 2015*).

## Hipposeq, a publicly available resource for hippocampal transcriptomics

All data presented here are accessible on the Hipposeq website (http://hipposeq.janelia.org), an interactive database that allows user-friendly analysis and visualization of gene expression data for individual genes, cohorts of genes, and entire transcriptomes. Through this site, data can be mined for *a priori* genes of interest, or alternatively investigated with discovery-based analysis tools. Raw and processed data are also available for download, enabling the user to export data into their own environment for more specialized analyses. This website and associated dataset expand upon and complement other existing publicly available gene expression databases, mostly notably the ABA (*Lein et al., 2007*). Our RNA-seq approach offers advantages that circumvent traditional issues associated with ISH; namely, providing class-specific data with a large dynamic range, helping to circumvent confounds associated with image-based analyses of gene expression that can be limited by

changes in cell density and/or labeling across sections. Of course, RNA-seq also has limitations relative to ISH, including an inherent lack of spatial information. In this way, the combination of Hippo-seq and the ABA ISH atlas provides a powerful set of tools that enables quantitative and histological whole genome insight into gene expression in the hippocampus.

## Materials and methods

Mice were housed on a 12-hr light/dark cycle with ad libitum food and water access. Experimental procedures were approved by the Institutional Animal Care and Use Committee at the Janelia Research Campus (protocol #14–118).

### Manual sorting, library preparation, and sequencing

All transgenic mice used (namely, granule cells from both blades of the DG: Rbp4-Cre KL100, mossy cells of the hilus: Lypd1-Cre NR151, CA3 and CA2 pyramidal cells: Mpp3-Cre KG118, CA1 pyramidal cells: Vipr2-Cre KE2) were generated by the Gene Expression Nervous System Atlas (GENSAT) project (*Gong et al., 2003*; *Gong et al., 2007*). Transgenic lines were maintained on a C57bl/6J background, with each line backcrossed at least one generation prior to use (note that strain-specific gene expression differences are likely minor [*Sandberg et al., 2000*]). Cre expression was reported by an Ai9 (tdTomato) mouse cross (*Madisen et al., 2010*), and double-positive mice of either sex were sacrificed (age P26-P35, either single- or group-housed) within a 3 hr time window approximately midway through the light cycle, with microdissection locations shown in *Figure 1—figure supplement 1*. In all cases, manual sorting to purify for fluorescent neurons from microdissected slices was performed according to previous methods (*Hempel et al., 2007*). For each cell class/region combination, three biological replicates (i.e., cell class/region from a different animal of the same genotype) with sufficient reproducibility (within-class Pearson correlation coefficient >0.90, a criterion determined after analysis) were obtained; biological replicates were re-obtained for datasets for correlations <0.90. This was the only exclusion criterion for datasets. No technical replicates were used in this study. Three biological replicates have been previously shown to be sufficient in detecting differences in gene expression known *a priori* (*Cembrowski et al., 2016*). On average, 112 ± 6 cells (mean ± SEM, n = 24 replicates) were recovered in the final purified pool for library preparation and sequencing.

Total RNA was isolated from each sample using PicoPure RNA Isolation kit (Life Technologies, Frederick, MD) including the on-column RNase-free DNase I treatment (Qiagen, Hilden, Germany) following the manufacturers' recommendations. Eluted RNA (11 ul) was dried in a speed vac to approximately 2–4 ul. ERCC control RNAs (Life Technologies) were added using 1 ul of 1:100,000 dilution for every 50 cells. cDNA was amplified from this input material using Ovation RNA-seq v2 kit (NuGEN, San Carlos, CA). Approximately half of the resulting cDNA was used to make the sequencing libraries using the Ovation Rapid DR Multiplexing kit (NuGEN). Four barcoded libraries were pooled per sequencing lane on a HiSeq 2500 (Illumina, San Diego, CA) and single-end 100 bp reads were generated. No randomization or blinding was used for sorting, library preparation, or sequencing.

### RNA-seq read alignment, quantification, differential expression, and analysis

Reads for each library (37.9 ± 1.3 million per replicate, n = 24 replicates) were mapped using TopHat v2.0.6 (http://ccb.jhu.edu/software/tophat/index.shtml) (*Trapnell et al., 2009*) against the mouse genome build NCBIM37 (mm9) combined with sequences corresponding to ERCC spike-in controls. The following options were used: '–num-threads 8 –GTF mouseGtf.gtf', where mouseGtf.gtf reflects the concatenated Ensembl NCBIM37 transcript annotation file and the annotated ERCC spike-in controls. With these settings, 77.5 ± 1.0% (29.6 ± 2.0 million per replicate, n = 24) of all reads aligned at least once to either the annotated transcriptome or genome.

After mapping, quantification and differential expression of the annotated mouseGtf.gtf was performed using Cuffdiff v2.1.1 (http://cole-trapnell-lab.github.io/cufflinks/) (*Trapnell et al., 2010*) using the accepted_hits.bam files for all replicates, with three biological replicates used for each dataset. The following options were used: "–frag-bias-correct mouseFa.fa –mask-file mouseMask.gtf –max-bundle-frags 10000000 –num-threads 8 –multi-read-correct –no-effective-length-correction", where

mouseFa.fa is the Ensembl NCBIM37 reference FASTA and mouseMask.gtf is a mask file that ignores all alignments corresponding to genes annotated in mouseGtf.gtf annotated as tRNA, rRNA, or snRNA. In addition, from inspecting gene tracks we also noticed that a few loci (namely, *Nat8l*, *Psd2*, *Xist*, *Gm15459*, and *Gm10335*) would occasionally produce many identical reads to one specific sequence with few or no alignment reads found elsewhere; these loci were also masked. Finally, when not explicitly examining ERCC controls (*Figure 1—figure supplement 2c*), ERCC 'loci' were also included in the mask. When considering spike-in controls, FPKM values were found to be highly reproducible across replicates ($r = 0.94 \pm 0.01$, $n = 24$ replicates, Pearson correlation coefficient).

The resulting data were analyzed in the R environment using a combination of cummeRbund v3.0 (http://compbio.mit.edu/cummeRbund/) (*Goff et al., 2013*) and custom scripts. General analysis conventions were as follows: a conventional threshold of FDR <0.05 was used for differential expression, allowing both under- and overexpressed genes to be identified (i.e., two-sided); a gene was considered $X$-fold enriched in a given region, relative to other regions, when the mean FPKM value was at least $X$-fold greater for all corresponding pairwise comparisons (e.g., for gene A to be $X$-fold enriched in dorsal CA1 relative to dorsal CA2 and dorsal CA3, $FPKM_{A,CA1dorsal} > X \cdot FPKM_{A,CA2dorsal}$ and $FPKM_{A,CA1dorsal} > X \cdot FPKM_{A,CA3dorsal}$); Pearson correlation coefficients were used to compare across datasets; error bars for FPKM values were taken from Cuffdiff's 95% CI model; and gene expression was required to obey FPKM>10 in at least one population to be included in differential expression or enriched population analyses. No randomization or blinding was used for computational handling of data. Raw and processed RNA-seq datasets were deposited in the National Center for Biotechnology Information (NCBI) Gene Expression Omnibus (GEO), accession number GSE74985, and analysis scripts can be downloaded from Github (https://github.com/cembrowskim/hipposeq.git).

For hierarchical clustering of datasets (*Figure 2a*), whole-transcriptome averaged FPKM values were processed by adding a pseudocount of 1 to all values, $\log_{10}$ transformed, and normalized on a sample-by-sample basis by the sample sum of the transformed FPKM value. The pairwise Jensen-Shannon distance was calculated across samples, and agglomerative clustering was performed on the distance matrix using complete linkage.

When plotting normalized FPKM heat maps (*Figures 2a*, *3a,b*, *4a,b*, *6e*, *7a,c,d*; Figure Supplements 1-2d, 3-1a,b,c, 5-1a), each gene (i.e., column) was normalized by dividing the expression values by the highest FPKM value across samples. When visualizing expression in single-gene bar plots (*Figures 2b*, *4c*, *5a,d,e*, *6b*; Figure Supplements 5-1b,c, 6–1), x-axis values are individual class/region datasets, such that filled bars represent dorsal samples, cross-hatched bars represented ventral samples, and coloring adheres to the convention of *Figure 1a*; y-axis values are gene expression values in FPKM.

For identifying marker genes corresponding to the clusters of *Figure 2a*, we searched for genes that were more than two fold enriched in every replicate of the desired sample(s) relative to the remaining sample(s). For identifying cell class-specific genes (*Figure 3a,b*; *Figure 4a,b*), we searched for genes that were more than 3-fold enriched on average in the desired population, relative to all other populations. For identifying regionally enriched granule cell marker genes with neuronal relevance (*Figure 5—figure supplement 1a*), we searched for genes that were more than 3-fold enriched at either of the two poles relative to the opposite pole.

For multidimensional scaling (MDS) of CA3, CA2, and CA1 dorsal datasets (*Figure 3d*), replicate FPKM values were processed by adding a pseudocount of 1 to all values, $\log_{10}$ transformed, and normalized on a sample-by-sample basis by the sample sum of the transformed FPKM value. The pairwise Jensen-Shannon distance was calculated across samples, and agglomerative clustering was performed on the distance matrix using complete linkage. Two-dimensional classical MDS was performed by using *cmdscale* in R with default arguments.

When examining the number of regionally enriched genes invariant to principal cell class (*Figure 6*), genes obeying FPKM>10 for each class/region combination were obtained ('expressed genes'), and from this list, for each dorsal-ventral comparison genes also being >2 fold enriched at either pole were identified ('enriched genes', *Figure 6a*). The overlap in enriched genes across respective regions was then determined (horizontal lines, *Figure 6c*). To derive chance levels for overlap, Monte Carlo simulations were performed. Here, genes were drawn at random without replacement from the expressed genes list from each dataset, with the total number of genes drawn equal to the number of enriched genes for each pairwise comparison. The ensuing results were

analyzed analogously to empirical data. This process was repeated 1000 times to characterize the chance distributions for all comparisons.

For comparing to MEC RNA-seq (*Figure 6—figure supplement 1*), we first identified all genes that were >2 enriched in each dorsal-ventral comparison for cell class in the trisynaptic loop (n=37). Next, we compared this list to those identified as differentially expressed (FDR < 5%) from RNA-seq on microdissected dorsal and ventral poles of the MEC (S7 *Dataset, Ramsden et al., 2015*). Genes present in this list that obeyed the same directionality of enrichment are highlighted in *Figure 6e* (dorsal: 6/12; ventral: 10/27).

WGCNA analysis (*Zhang and Horvath, 2005*) was performed (*Figure 7*) by first identifying the 1000 most variable genes across datasets (FPKM$_{MIN}$=5). The correlation matrix $C$ of these genes was subsequently obtained according to $C_{ij} = r_{ij}$, where $r_{ij}$ was the Pearson correlation coefficient of genes $i$ and $j$ across datasets. The dissimilarity matrix $D$ was then calculated by taking $D_{ij} = 1 - \left(\frac{C_{ij}+1}{2}\right)$. Divisive hierarchical clustering was then performed on $D$ by the *diana* method in R with default parameters, and modules were obtained by choosing a cut height of 10 on the computed dendrogram. Genes associated with each cohort were analyzed by DAVID (Database for Annotation, Visualization and Integrated Discovery) (*Huang et al., 2009a*, *2009b*). The default Gene Ontology terms (GOTERM_BP_FAT, GOTERM_CC_FAT, GOTERM_MF_FAT) (*Ashburner et al., 2000*) and KEGG (Kyoto Encyclopedia of Genes and Genomes) pathways (*Kanehisa and Goto, 2000*; *Kanehisa et al., 2014*) were analyzed by the Functional Annotation Chart, and terms that obeyed a Benjamini (adjusted) p-value <0.05 were considered for further analysis.

## Cross-validation of alignment, quantification, and differential expression of RNA-seq data

First, to cross validate results of the TuxedoSuite pipeline with alternative quantification and differential expression software, we used HTSeq to quantify expression in a count-based fashion (*Figure 1—figure supplement 3a–g*). For each sample, the mapped reads from TopHat were quantified by HTSeq using htseq-count with the previous mouseGtf.gtf and the following options: "–format=bam –stranded=no". After quantification, the count data for individual samples were merged into one file, and analyzed by DESeq2 and custom scripts in the R environment. Values are reported in counts per million (CPM), and a CPM cutoff of 20 was used for fold change analysis, which retained a similar number of genes to FPKM=10 threshold used elsewhere.

To examine the choice of alignment software on quantification and differential expression (*Figure 1—figure supplement 3h–j*), in a second set of analysis we first aligned reads with STAR v2.5.1a (*Dobin et al., 2013*) rather than TopHat. For each sample, reads were mapped according to "–runThreadN 8 –genomeDir starGenomeDir –outSAMtype BAM SortedByCoordinate", where starGenomeDir was the directory containing the STAR genome index files. Output BAM files were then processed for quantification and differential expression according to the Cuffdiff approach described above.

## Analysis of ABA ISH database

When cross-validating the results of RNA-seq, we examined coronal ISH images from the ABA (*Lein et al., 2007*) (except for *Figure 6d*, where sagittal sections were used to visualize dorsal and ventral trisynaptic loops in the same section). To validate genes identified by RNA-seq as enriched in hierarchical clustering subgroups (*Figure 2a*), ISH expression profiles at the corresponding dorsal or ventral section were examined, and the validation was counted as a success if there was obvious expression by eye in the enriched subgroup (occurring in ~81% of cases; 124/153 genes, *Supplementary file 1*). Similar approaches were used for mossy cell marker genes (*Figures 3b* and *4b*, *Supplementary file 2*). Expression in all representative ISH images shown in the text was consistent with other sections near the same anterior-posterior location in the same animal, as well as with at least one additional animal in the ABA (with the exception of *Nr2f2* (*Figure 5e*), which had ubiquitous labeling in sagittal sections, inconsistent with the coronal expression pattern employing a different animal and probe, as well as *Inf2* (*Figure 6—figure supplement 1*), which had only one animal).

To examine reproducibility of RNA-seq vs. ISH cross-validation, two additional observers were independently shown a randomly chosen subset of images corresponding to enriched genes of *Figure 2a* and asked to identify the cell class(es) that exhibited expression. This scoring was

performed blind to the RNA-seq result and incorporated negative control images wherein RNA-seq expression was not cell class-specific. This blind, outside-observer assessment correctly identified the enriched populations at a similar success rate (77% and 88% of genes for two independent observers, with n = 20/26 and 23/26 randomly selected genes correctly identified respectively).

## Fluorescence imaging

Images of large regions of tissue (i.e., complete dorsal and ventral CA1) were acquired on a whole-slide digital scanner (Pannoramic 250 Flash, Perkin Elmer, Waltham, PA) using a 20x objective. Cellular resolution images were acquired with a confocal microscope (LSM 710 Carl Zeiss Microscopy, Jena, Germany) using a 20x objective. Some images were post-processed in Fiji, including pseudo-coloring to adhere to the coloring conventions of different cell classes.

## Acknowledgements

The authors would like to thank Brett Mensh, Erik Bloss, Julia Bachman, William Kath, and other members of the Spruston laboratory for helpful discussions, Deanna Otstot for help with breeding, Jody Clements and Jonathan Epstein for website construction, and Andrew Lemire and Serge Picard for helping to coordinate and generate RNA-seq data. We are also grateful to the Allen Institute for Brain Science for providing the public Allen Mouse Brain Atlas (http://mouse.brain-map.org/), and the creators of TuxedoSuite, HTSeq, and DESeq2 for providing essential, publicly available RNA-seq analysis and visualization software. Funding was provided by the Howard Hughes Medical Institute.

## Additional information

### Funding

| Funder | Author |
| --- | --- |
| Howard Hughes Medical Institute | Mark S Cembrowski<br>Lihua Wang<br>Ken Sugino<br>Brenda C Shields<br>Nelson Spruston |

The funders had no role in study design, data collection and interpretation, or the decision to submit the work for publication.

### Author contributions

MSC, Conception and design, Analysis and interpretation of data, Drafting or revising the article; LW, KS, BCS, Conception and design, Acquisition of data; NS, Conception and design, Drafting or revising the article

### Author ORCIDs

Nelson Spruston, http://orcid.org/0000-0003-3118-1636

### Ethics

Animal experimentation: Experimental procedures were approved by the Institutional Animal Care and Use Committee at the Janelia Research Campus (protocol #14-118).

## Additional files

**Supplementary files**

• Supplementary file 1. Dendrogram marker genes.

• Supplementary file 2. Marker genes for dentate gyrus mossy cells.

• Supplementary file 3. Dorsal and ventral dentate gyrus granule cell marker genes.

• Supplementary file 4. List of Allen Mouse Brain Atlas images shown in text

### Major datasets

The following dataset was generated:

| Author(s) | Year | Dataset title | Dataset URL | Database, license, and accessibility information |
|---|---|---|---|---|
| Cembrowski M, Spruston N | 2016 | Hipposeq: an RNA-seq based atlas of gene expression in excitatory hippocampal neurons | http://www.ncbi.nlm.nih.gov/geo/query/acc.cgi?token=adsveykeprejbe-j&acc=GSE74985 | Publicly available at NCBI Gene Expression Omnibus (accession no. GSE74985) |

The following previously published dataset was used:

| Author(s) | Year | Dataset title | Dataset URL | Database, license, and accessibility information |
|---|---|---|---|---|
| Cembrowski M, Spruston N | 2016 | Spatial gene expression gradients underlie prominent heterogeneity of CA1 pyramidal neurons | http://www.ncbi.nlm.nih.gov/geo/query/acc.cgi?acc=GSE67403 | Publicly available at NCBI Gene Expression Omnibus (accession no. GSE67403) |

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
