## [Decision Letter]

Thank you for submitting your article "Hipposeq: a comprehensive RNA-seq database of gene expression in hippocampal principal neurons" for consideration by *eLife*. Your article has been reviewed by 2 peer reviewers, and the evaluation has been overseen by Eve Marder as the Senior Editor.

The reviewers have discussed the reviews with one another and the Reviewing Editor has drafted this decision to help you prepare a revised submission.

1) Please note that the most important issue that arose during the review is that the reviewers did not have access to the web site with the actual data nor to the analysis code. As you know, *eLife* will expect you to make both of these public, and it seems reasonable that the reviewers should have access to these in order to make their final judgments of the work. Consequently, along with preparing a revision to be responsive to the issues raised below, please ensure that these are available to us at the next submission. One of the reviewers requested that you use GitHub (or equivalent) rather than FigShare.

Summary:

This manuscript describes gene expression datasets for several classes of excitatory hippocampal neuron: CA1 (dorsal and ventral), CA2 (dorsal) and CA3 pyramidal cells (dorsal and ventral), DG granule cells (dorsal and ventral) and DG mossy cells (dorsal). The datasets appear to have been carefully generated and analysed, and the results and methods are clearly described. It is likely the datasets will be useful as a "ground truth" of population level gene expression profiles of each of the investigated cell types.

Points for revision:

1) The study claims to have characterised gene expression for every excitatory neuronal class in the hippocampus. While we agree the major classes have been characterised, there are additional excitatory cell classes that were not investigated, e.g. semilunar granule cells in the dentate gyrus (Williams et al., J. Neurosci. 2007; Larimer and Strowbridge, Nature Neuroscience 2010), radiatum giant cells in CA1 (Gulyas et al., Eur. J. Neurosci. 1998; Bullis et al., J. Physiol. 2007), and CA3 granule cells (Szabadics et al., J. Neurosci. 2010). These classes have distinct morphology and biophysical properties. It would not be surprising if their gene expression profiles are also distinct. Given the good evidence for these additional excitatory cell classes, the claims for completeness should be qualified and the excitatory cell classes not characterised should be acknowledged.

2) The replicate correlation score is much lower for granule cells. Is there a known reason for this? Were granule cells sampled from upper or lower blade of the DG? Or both? Is there a difference?

3) Mapping of reads used TopHat. For many applications this has been superseded by STAR (Dobin et al., Bioinformatics 2013) which has been suggested to map more accurately. It might be worth checking whether this gives improvements sufficient to reveal any additional differentially expressed genes.

4) The Hipposeq resource is referred to multiple times in the manuscript, but is not accessible so it is not possible to comment on the "resource" aspect of the manuscript. It's also not clear what the resource provides that could not be achieved by downloading the data via GEO and using standard analysis tools. Ease of use? Novel analyses?

5) Materials and methods. '12-hour light/dark cycle'. At what time were the animals sacrificed? If the time differed between animals, does this contribute to any variance in the data?

6) In the last paragraph of the subsection “Generating a cell- and region-specific RNA-seq database for the hippocampus”. How many cells (mean ± SD)? I think this is in the Methods, but would be helpful to make clear in the Results section.

7) In the first paragraph of the subsection “Manual sorting, library preparation, and sequencing”. Which C57bl/6 strain? If not generated on this background, for how many generations are the lines back-crossed?

8) How was the integrity of the RNA assessed? E.g. was there a threshold RIN value?

9) There is increasing evidence for diversity of CA1 along the radial (deep vs. superficial) axis. Please consider.

10) The figure legends are a bit sparse and could benefit from adding a more detailed description of the data.

11) Figure 1—figure supplement 1. Please label the various panels with the identity of the transgenic line.

12) Although some mouse lines do show fairly specific expression limited to specific subregions, other lines show very broad patterns of expression, particularly for CA2 (Figure 1—figure supplement 1). How did the authors manage to get a CA2-specific population to examine?

13) Did the authors look for inhibitory specific cell markers (e.g. GAD65,67) to judge the level of contamination from inhibitory neurons?

14) Please give a better description of the bar graphs in Figure 2, Figure 4, etc. Is the x-axis the cell type? If so, it would be helpful to indicate this on the panels and in the legend. Also, what is the y-axis? FPKM? Please define. The normalized color plots of FPKM are hard to understand, and the description in the Methods are obscure for a non-expert. What is the being normalized? The bar-graph quantification much more useful than the normalized color plots. Perhaps the authors may wish to include a table in the supplemental material listing actual fold differences in expression of various genes?

---

## [Author Response]

*1) Please note that the most important issue that arose during the review is that the reviewers did not have access to the web site with the actual data nor to the analysis code. As you know, eLife will expect you to make both of these public, and it seems reasonable that the reviewers should have access to these in order to make their final judgments of the work. Consequently, along with preparing a revision to be responsive to the issues raised below, please ensure that these are available to us at the next submission.*

Our website, http://hipposeq.janelia.org, is currently available for external consideration, with the username *private* and the password *forreviewers*. This information was provided in the cover letter of our original submission. To ensure that reviewers don’t miss it, we have now added the credentials to the main text as well; password protection and the associated main text credential references will be removed before publication.

*One of the reviewers requested that you use GitHub (or equivalent) rather than FigShare.* In addition to Figshare, our code is now also available through Github at https://github.com/cembrowskim/hipposeq.git, and noted as such in the manuscript.

*Points for revision: 1) The study claims to have characterised gene expression for every excitatory neuronal class in the hippocampus. While we agree the major classes have been characterised, there are additional excitatory cell classes that were not investigated, e.g. semilunar granule cells in the dentate gyrus (Williams et al., J. Neurosci. 2007; Larimer and Strowbridge, Nature Neuroscience 2010), radiatum giant cells in CA1 (Gulyas et al., Eur. J. Neurosci. 1998; Bullis et al., J. Physiol. 2007), CA3 granule cells (Szabadics et al., J. Neurosci. 2010). These classes have distinct morphology and biophysical properties. It would not be surprising if their gene expression profiles are also distinct. Given the good evidence for these additional excitatory cell classes, the claims for completeness should be qualified and the excitatory cell classes not characterised should be acknowledged.* The reviewer makes an excellent point. As such, we have removed all claims of profiling every excitatory neuronal class in the hippocampus, instead emphasizing that we have investigated the major neuronal classes. In addition, we have explicitly indicated in the Discussion that there are additional classes that require further investigation, specifically highlighting the classes the reviewer points out.

*2) The replicate correlation score is much lower for granule cells. Is there a known reason for this?*

The relatively low correlation in DG granule cells (dorsal in particular) emerges largely due to one dorsal replicate having a slightly lower correlation with the other two dorsal DG granule cell replicates (*r* = 0.94 and 0.94 when correlating this replicate to the other two dorsal DG replicates). We do not know the exact reason for this lower correlation, but have ruled out contributions from off-target effects (Figure 1—figure supplement 2).

We also hasten to emphasize that the overall dorsal DG granule cell correlation is still very high (~95% correlation), and seems relatively low in Figure 1—figure supplement 2 due to the near- perfect correlations found in our other datasets (typically 98-99% correlated).

*Were granule cells sampled from upper or lower blade of the DG? Or both?*

Cells for sorting were taken from both the upper and lower blades of the DG. We have noted this in the revised manuscript.

*Is there a difference?*

We were also interested in whether there were discernable differences in gene expression between the upper and lower blades of the dentate gyrus, which we have previously investigated by examining the Allen Brain Atlas in situhybridization database. However, neither manual searching nor automated methods were able to identify clear gene expression differences between the upper and lower blades. We have noted this in the revised manuscript.

*3) Mapping of reads used TopHat. For many applications this has been superseded by STAR (Dobin et al., Bioinformatics 2013) which has been suggested to map more accurately. It might be worth checking whether this gives improvements sufficient to reveal any additional differentially expressed genes.* The reviewer makes an excellent suggestion. We have now conducted this analysis, which produced few differences relative to our previously used pipeline, and note this in the newest version of the manuscript (Figure 1—figure supplement 3).

To elaborate on this approach, we used STAR v2.5.1a with default arguments to conduct alignment, and subsequently performed quantification and differential expression with Cuffdiff. This approach was analogous to our previous pipeline, with the sole exception of changing the alignment software (STAR vs. TopHat), allowing us to assess the choice of software on downstream quantification and differential expression.

First, we examined whether the transcriptomes were quantitatively similar across alignment approaches. Comparing average transcriptomes across alignment software, we found that most genes had similar FPKM values between the two approaches. For example, for dorsal CA1, the Pearson correlation between the outputs of the two pipelines was 0.98 (shown in Figure 9; for all datasets, Pearson correlation = 0.98 ± 0.00, mean ± SD, n=8 datasets).

Author response image 1.**DOI:**
http://dx.doi.org/10.7554/eLife.14997.021

We note that there are indeed differences between the two aligners for some genes, which are expected due to some fundamentally different properties of the two aligners (e.g., relativetolerances for mismatches and splice junctions; see Engström et al., Nature Methods, 2013 for detailed analysis). However, the vast majority of genes were found to be expressed to similar levels between the two alignment strategies, indicating that to a first approximation the overall measured transcriptomes were robust to alignment software.

As requested, we next investigated the robustness of differential expression calls across alignment software. Here, too, the results were similar between the two aligners; e.g., for dorsal vs. ventral CA1, the overall extent of differential expression was nearly identical, with almost all genes being shared between the two alignment schemes (Figure 10, with differentially expressed genes highlighted in green; 955/1015 = 94% of differentially expressed genes found by Tophat were also identified by STAR).

Author response image 2.**DOI:**
http://dx.doi.org/10.7554/eLife.14997.022

These results were similar for differential expression analyses across all pairwise comparisons (95.0 ± 1.3% of differentially expressed genes found by TopHat approach were shared with STAR, with STAR identifying 6.8 ± 1.3% more genes than TopHat; mean ± SD, n = 28 pairwise comparisons for each). This revealed that STAR did not make a substantial increase in differential expression calls relative to TopHat, and thus reinforces the robustness of our results to the particular choice of alignment software.

*4) The Hipposeq resource is referred to multiple times in the manuscript, but is not accessible so it is not possible to comment on the "resource" aspect of the manuscript.*

The website is now accessible with the credentials provided in response to the comments above.

*It's also not clear what the resource provides that could not be achieved by downloading the data via GEO and using standard analysis tools. Ease of use? Novel analyses?*

Primarily, our website will be of use to individuals and laboratories that do not have prior knowledge in the statistical analysis and/or visualization of RNA-seq data. For this audience, Hipposeq provides a readily available interface that does not require coding experience or statistical knowledge to quickly arrive at mathematically well-principled and biologically informative results.

Even for individuals and laboratories that have background in the bioinformatics of RNA-seq analysis, moving from raw data to finalized conclusions can be a challenging and time-consuming task (e.g., Garber et al., Nature Methods, 2011). We have worked hard to provide a suite of analysis and visualization tools that allow our RNA-seq data to be immediately examined and interpreted. In particular, this suite includes exploratory analysis tools that interactively facilitate moving from broad analyses (e.g., comparing transcriptomes) to more refined, low-level searches (e.g., comparing gene cohorts or individual genes). These readily available analysis tools will a helpful asset for novel and experienced users alike.

*5) Materials and methods. '12-hour light/dark cycle'. At what time were the animals sacrificed? If the time differed between animals, does this contribute to any variance in the data?* To avoid any contributions arising from circadian rhythms, all animals in this study were sacrificed within a 3-hour time window approximately halfway through the light cycle. We now explicitly mention this in the Methods of the revised manuscript.

*6) In the last paragraph of the subsection “Generating a cell- and region-specific RNA-seq database for the hippocampus”. How many cells (mean ± SD)? I think this is in the Methods, but would be helpful to make clear in the Results section.* This value (112 ± 6 cells per biological replicate, n = 24 replicates) is now included in the Results section.

*7) In the first paragraph of the subsection “Manual sorting, library preparation, and sequencing”. Which C57bl/6 strain?*

C57bl/6J strain was used; this is now included in the manuscript.

*If not generated on this background, for how many generations are the lines back-crossed?*

Most transgenic lines used in this study were not generated on the C57bl/6 background, but all lines were bred to C57bl/6 at least one generation, which we now note in the Methods. We do not believe that the number of generations of the backcross has a dramatic effect on the observed results, for three reasons. First, our RNA-seq results were cross-validated by ABA ISH data, which was performed on a C57Bl6/J background. Second, we have previously shown (Cembrowski et al., Neuron, 2016) that our RNA-seq results recapitulate differential expression results from other mouse species (e.g., cf. 129SvEv mice from Leonardo et al., Neuroscience, 2006). Third, consistent with the previous two points, gene expression differences across mouse strains have been shown to be negligible (e.g., <1% of expressed genes in the hippocampus are differentially expressed between 129SvEv and C57bl/6 lines, Sandberg et al., PNAS, 2000).

*8) How was the integrity of the RNA assessed? E.g. was there a threshold RIN value?* As we have relatively few cells per replicate, we have insufficient RNA to determine RIN scores. However, we have two lines of argument that suggest our RNA integrity is nevertheless very high. First, our preparation keeps neurons alive as they go into lysis buffer containing DTT, preserving RNA integrity at –80C until downstream RNA extraction and library preparation.

Second, the high correlation found between RNA-seq replicates suggests that our RNA integrity is high, as poor RNA healthy tends to produce poorly correlated replicates.

*9) There is increasing evidence for diversity of CA1 along the radial (deep vs. superficial) axis. Please consider.* We are very interested in this as well, and investigated it in depth in a recent publication (Cembrowski et al., Neuron, 2016). To summarize these findings, we found that the superficial-deep axis does contain transcriptional differences; however, these differences were quantitatively much smaller than dorsal-ventral differences in CA1 (see Figure S3, Cembrowski et al., Neuron, 2016).

*10) The figure legends are a bit sparse and could benefit from adding a more detailed description of the data.* We have incorporated additional details to the legends, especially elaborating on the conventions of bar graphs and heat maps used throughout this paper (as requested in point 14 below).

*11) Figure 1—figure supplement 1. Please label the various panels with the identity of the transgenic line.* Done.

*12) Although some mouse lines do show fairly specific expression limited to specific subregions, other lines show very broad patterns of expression, particularly for CA2 (Figure 1—figure supplement 1). How did the authors manage to get a CA2-specific population to examine?* The transgenic animals used for characterization of CA2 (Mpp3-cre x Ai9) showed a sharp drop in fluorescence at the CA1-CA2 border. Using this border as a landmark, we carefully microdissected out a region immediately before this drop in fluorescent expression, postulating that this was a small region corresponding to CA2 pyramidal cells. After obtaining transcriptomes for this labeled population, we verified post hoc the enrichment of genes previously associated with CA2 pyramidal cells (indicating CA2 was effectively targeted; e.g., Figure 3) and the depletion of previously identified genes associated with CA3 and CA1 pyramidal cells (indicating off-target CA3 and CA1 pyramidal cells were effectively excluded; e.g., Figure 2).

*13) Did the authors look for inhibitory specific cell markers (e.g. GAD65,67) to judge the level of contamination from inhibitory neurons?* Yes. These data are part of Figure 1—figure supplement 2, and show that all replicates are devoid of expression of gene cohorts associated with a variety of off-target cells, including interneurons.

*14) Please give a better description of the bar graphs in Figure 2, Figure 4, etc. Is the X axis the cell type, if so, it would be helpful to indicate this on the panels and in the legend? Also, what is the Y axis? FPKM? Please define.*

These conventions are now explained in the Methods, as well as in the figure legend corresponding to the first bar graph to appear in the manuscript (Figure 2).

*The normalized color plots of FPKM are hard to understand, and the description in the Methods are obscure for a non-expert. What is the being normalized?*

Again, these conventions are now explained in the Methods, as well as incorporated into the figure legend for the first heat map to appear in the manuscript (Figure 2).

*The bar-graph quantification much more useful than the normalized color plots.*

We certainly agree that the bar graphs are much easier to interpret than the heat maps when considering small numbers of genes. However, heat maps are the easiest way to quickly visualize expression for more comprehensive lists of genes, and a standard visualization technique for representation of gene cohort data in genomics (e.g., Sugino et al., Nat. Neuro., 2006; Bernard et al., Neuron, 2012; Zhang et al., J. Neurosci., 2014). As such, we have used bar plots when considering individual genes or small groups of genes, and heat maps when trying to communicate the full extent of gene expression differences between samples.

Perhaps the authors may wish to include a table in the supplemental material listing actual fold differences in expression of various genes?

We have provided an interface on the website to allow users to select desired populations and parameters for enrichment assays, and also included a table listing fold differences for DG granule cells ([Supplementary-material SD3-data]), which were specifically examined and referenced in the manuscript.